# Preventive Counseling in Routine Prenatal Care—A Qualitative Study of Pregnant Women’s Perspectives on a Lifestyle Intervention, Contrasted with the Experiences of Healthcare Providers

**DOI:** 10.3390/ijerph19106122

**Published:** 2022-05-18

**Authors:** Laura Lorenz, Franziska Krebs, Farah Nawabi, Adrienne Alayli, Stephanie Stock

**Affiliations:** Institute of Health Economics and Clinical Epidemiology (IGKE), Faculty of Medicine and University Hospital Cologne, University of Cologne, 50935 Cologne, Germany; franziska.krebs@uk-koeln.de (F.K.); farah.nawabi@uk-koeln.de (F.N.); adrienne.alayli@uk-koeln.de (A.A.); stephanie.stock@uk-koeln.de (S.S.)

**Keywords:** patient experience, prevention, qualitative research, pregnancy, gestational weight gain, maternal health, lifestyle intervention

## Abstract

Maternal lifestyle during pregnancy and excessive gestational weight gain can influence maternal and infant short and long-term health. As part of the GeMuKi intervention, gynecologists and midwives provide lifestyle counseling to pregnant women during routine check-up visits. This study aims to understand the needs and experiences of participating pregnant women and to what extent their perspectives correspond to the experiences of healthcare providers. Semi-structured interviews were conducted with 12 pregnant women and 13 multi-professional healthcare providers, and were analyzed using qualitative content analysis. All interviewees rated routine check-up visits as a good setting in which to focus on lifestyle topics. Women in their first pregnancies had a great need to talk about lifestyle topics. None of the participants were aware of the link between gestational weight gain and maternal and infant health. The healthcare providers interviewed attributed varying relevance regarding the issue of weight gain and, accordingly, provided inconsistent counseling. The pregnant women expressed dissatisfaction regarding the multi-professional collaboration. The results demonstrate a need for strategies to improve multi-professional collaboration. In addition, health care providers should be trained to use sensitive techniques to inform pregnant women about the link between gestational weight gain and maternal and infant health.

## 1. Introduction

Overweight and obesity are major public health challenges and risk factors for subsequent diseases in both children and adults [1,2]. The foundations for overweight and obesity are established early in life. There is growing evidence that excessive gestational weight gain and the maternal lifestyle during pregnancy can influence the child’s risk of obesity and chronic disease in the long term [3,4,5]. Furthermore, excessive gestational weight gain is a risk factor for pregnancy and birth complications, such as preeclampsia, macrosomia, cesarean section, gestational diabetes mellitus (GDM), and Large for Gestational Age (LGA) [3,4,6,7,8,9,10,11,12].

Due to this, pregnancy is described as a unique “window of opportunity” for preventive interventions aimed at improving maternal and child health [13]. Modifiable behavioral risk factors for adverse pregnancy outcomes and lifelong non-communicable diseases include a lack of physical activity, unhealthy diet, alcohol consumption, and smoking during pregnancy [14]. Even though adopting a healthy lifestyle before pregnancy is beneficial for the health of the mother and child [15,16], the period of pregnancy is discussed as a “teachable moment” and may, therefore, be a favorable time for interventions. This is because pregnant women may be particularly motivated toward ensuring that they are in good health, and the importance of risk factor modification and healthy lifestyles can be reinforced effectively [17,18]. There is evidence that lifestyle interventions can be effective in improving maternal lifestyle and limiting excessive gestational weight gain [14,19,20,21,22,23].

The percentage of women experiencing excessive weight gain during pregnancy based on National Academy of Medicine (NAM; formerly known as the Institute of Medicine, IOM) guidelines [24] ranges from 47 to 68.5% across various studies and countries [3,7,10,25,26,27,28]. These figures highlight the urgent need for preventive intervention. The International Weight Management in Pregnancy (i–WIP) Collaborative Network published a “statement on tackling obesity in pregnancy”, in which it called for the incorporation of lifestyle counseling into routine prenatal care [29].

In Germany, lifestyle topics are not discussed consistently in the context of prenatal care [30,31]. Prenatal care in Germany is provided by office-based gynecologists and midwives, and focuses mainly on the early identification of diseases and developmental problems in the fetus [30,31]. While prenatal care can, in principle, be provided by midwives and gynecologists individually, it should preferably be administered in a complementary manner [31]. Almost all pregnant women in Germany utilize prenatal screening appointments, which are paid for by the Statutory Health Insurance. As a result, they are monitored closely throughout the entire course of their pregnancies [31]. In addition to this, gynecologists are often the main healthcare providers (HCPs) for women of childbearing age and accompany these women for many years during regular preventive check-ups [32]. As such, routine prenatal care provides an ideal setting for lifestyle intervention. The GeMuKi intervention (acronym for “Gemeinsam gesund: Vorsorge plus für Mutter und Kind”—Strengthening health promotion: enhanced check-up visits for mother and child), carried out in Germany, uses this setting to address lifestyle topics and to involve multiple HCPs who consistently complement each other [33,34].

In order for lifestyle interventions to be effective and sustainable, they must be adapted to the needs of pregnant women. At the same time, HCPs who implement these in routine care need to find the interventions acceptable and feasible [35]. A qualitative study conducted in the U.S. showed that most women had a positive attitude toward counseling during pregnancy, while HCPs discussed barriers to counseling, including, among others, a lack of time, lack of patient interest, or inadequate training [36]. A German study revealed information gaps among pregnant women in the fields of healthy eating and weight gain, as well as the need for information and motivation regarding suitable forms of exercise during pregnancy [37]. As demonstrated by an integrative review, evidence regarding women’s overall experience with regard to prenatal care is currently limited and further research is needed to enable HCPs to modify their care to more adequately fit women’s needs [38].

In light of this, this study aims to answer the following research questions: What needs, demands, and experiences do women have with regard to the preventive lifestyle counseling provided in the GeMuKi intervention? How do their perspectives correspond to the experiences of HCPs? The results can be used to develop strategies for adapting and improving prenatal care service structures.

## 2. Materials and Methods

### 2.1. Backrgound of This Study: The GeMuKi Intervention

This qualitative study was conducted as part of the process evaluation of the GeMuKi trial. The GeMuKi trial implemented a computer-assisted multi-professional intervention in order to address the lifestyle-related risk factors for overweight and obesity in expecting mothers and their infants. The intervention was carried out in five intervention regions of the southern German state of Baden-Wuerttemberg between January 2019 and January 2022 [33,34].

Embedded into regular check-up visits during pregnancy, six additional preventive counseling sessions were provided: four by trained gynecologists and two by trained midwives. All HCPs who delivered the intervention received eight hours of training in advance on lifestyle topics and on motivational interviewing (MI) techniques. MI is a client-centered approach designed to evoke intrinsic motivation for behavioral change [39,40]. The counseling topics were based on the national recommendations for a healthy lifestyle during pregnancy issued by the ‘Healthy Start—Young Family Network’ (“Netzwerk Gesund ins Leben”) [41]. During each counseling session, the women were asked to choose from the following topics: nutrition, water intake, physical activity, breastfeeding, alcohol, nicotine, and drug use. At the end of each session, the women and HCPs agreed on jointly set SMART (Specific, Measurable, Achievable, Reasonable, Time-Bound) goals for lifestyle changes. The achievement of these goals was then discussed in the next counseling session. The GeMuKi intervention included a novel shared telehealth platform that aids multi-professional HCPs during the counseling process (the GeMuKi-Assist counseling tool) and a corresponding app (the GeMuKi-Assist app) for the women participating in the intervention. One of the features used allowed HCPs to enter each women’s jointly agreed SMART goals into the GeMuKi-Assist counseling tool. After each counseling session, the participants received a reminder (push notification) of their lifestyle goals in their GeMuKi-Assist app. Further details on the GeMuKi trial and the GeMuKi intervention can be found elsewhere [33,34,42]. The GeMuKi trial was designed as a hybrid effectiveness–implementation trial, meaning that data on effectiveness and implementation were collected simultaneously [43]. The results on the effectiveness of the intervention, which was evaluated using a cluster randomized controlled design, are yet to be published.

### 2.2. Study Design

The report and conduct of this study are based on the ‘COnsolidated criteria for REporting Qualitative research’ (COREQ) (Appendix A) [44].

Qualitative interviews were conducted alongside the GeMuKi trial as part of the process evaluation during the first year of implementation. In order to answer the research question, an in-depth perspective from both the participating pregnant women and the HCPs was required. The use of qualitative methods appeared to be most appropriate, since this allowed an intensive description of the needs and perceptions of the interviewees.

Ethical approval was obtained from the University Hospital of Cologne Research Ethics committee on 22 June 2018 (ID: 18-163) and from the State Chamber of Physicians in Baden-Wuerttemberg on 28 November 2018 (ID: B-F-2018-100).

The interviews were conducted using semi-structured interview guides, which can be found in the Appendix A (Appendix A). To systematize the research interest, the development of the interview guides was informed by theoretical frameworks for the factors that influence implementation. The frameworks included were the ‘Implementation outcomes’ developed by Proctor et al. 2011 [45] and the ‘Tailored Implementation for Chronic Diseases (TICD) checklist’ [46], which is based on a synthesis of frameworks and taxonomies for determinants of professional practice. The interview guides contain open-ended questions regarding the procedure and the topics of the counseling sessions, as well as the participants’ satisfaction with the intervention and the needs of the pregnant women and HCPs. Depending on the flow of the conversation, the open-ended questions allowed individuals to bring up topics not covered by the interview guides.

At the end of the interviews, once the closing question had been answered, the pregnant women were asked to answer some questions related to sociodemographic factors and their pregnancy, while HCPs were asked about their professional experience and working environments. The interview guides were tested and discussed with women of childbearing age, experts from professional associations of gynecologists and midwives, and the project’s scientific advisory board.

### 2.3. Recruitment and Sample

The sample for this study was drawn from women and HCPs who were enrolled in the GeMuKi-trial. HCPs and pregnant women were invited to participate if they had undergone at least two counseling sessions. This applied to 23 gynecologists and their medical assistants from 17 gynecologic practices, 7 midwives, and 59 pregnant women. Pregnant women, gynecologists, and medical assistants were invited by postal mail to participate in the interviews. Letters of invitation were sent out to the women in June 2019, while invitations to the gynecological practices were sent out in October 2019 (in one of the five regions, the recruitment of interviewees was carried out one year later, as the implementation of the intervention in this region started one year later. This involved only one pregnant woman and two medical assistants). Midwives were recruited exclusively via telephone calls in October 2019 due to their limited postal accessibility.

Only two pregnant women and one medical assistant accepted the invitation, while two gynecologists and one medical assistant declined. No response was received from the remaining invitees. Because of this, all of the remaining participants already invited were contacted successively again by phone to ask if they were interested in an interview. While all contacted pregnant women were willing to be interviewed, 18 of the eligible gynecologists and 4 of the eligible midwives either rejected participation due to a lack of time or could not been reached. An appointment was scheduled with all of those who were interested. Once the interview was over, all of the interviewees received a gift (voucher) worth 15–20 euros as a thank you for their participation. After 12 interviews had been conducted with pregnant women, data saturation was discussed by the research team as no new themes emerged in the interviews. This was not possible in the same way for the HCP interviews, as no more HCPs could be recruited for an interview. The final sample consisted of 25 interviewees, of whom 12 were pregnant women and 13 were multi-professional HCPs (five gynecologists, five medical assistants, and three midwives). The sample characteristics are displayed in Table 1 and Table 2. The participating women were about 33 years old on average, had an average body mass index (BMI) of 25.6, and half of them were first-time mothers. All of the interviews were conducted in the last trimester of pregnancy. The interviewed HCPs were mostly female, and their level of professional experience varied greatly between 4 and 42 years. They all had between 8 and 12 months of experience in implementing the GeMuKi intervention.

### 2.4. Data Collection

The data collection for this study took place between July 2019 and March 2020 (in one of the five regions, the interviews were carried out in October and November 2020, as the implementation of the intervention in this region started one year later. This involved only the interviews with one pregnant woman and two medical assistants. These interviews were conducted during the COVID-19 pandemic. As the GeMuKi-intervention and the interviews for this study could be carried out in the same way as before the pandemic, there were no substantial differences). The first author (L.L.; female), who is a sociologist by training and an experienced qualitative researcher conducted 25 qualitative interviews. The interviewer was part of the evaluation team and had not met the interviewees before. The interviewees were informed in advance that the interviews would discuss their personal perspectives on and experiences of prevention and lifestyle counseling in prenatal care. They knew that their insights were needed to understand if the intervention fit their expectations and to improve the implementation process of the intervention in case of a national rollout. The interviews with the gynecologists were conducted in person in their offices. The interviews with the pregnant women, midwives, and medical assistants were conducted via telephone. All of the interviews were recorded digitally, anonymized, and transcribed verbatim according to the rules published by Dresing/Pehl (2011) [47]. The interviews with the pregnant women took an average of 21 min. The interviews with the medical assistants lasted a similar amount of time (17 min), whereas the interviews with the midwives and gynecologists took longer (Table 1 and Table 2). A second researcher (F.K. or F.N.) was present during the interview and documented the atmosphere and specifics during the interview in a postscript. They also made sure that all of the aspects of the interview guide were covered.

### 2.5. Data Analysis

The transcribed interviews were analyzed by two researchers using ‘thematic qualitative text analysis’ as described by Kuckartz (2014), a particular form of qualitative content analysis [48,49]. An inductive–deductive category-based approach was used [48]. L.L. developed the category system. Initially, only deductive categories derived from the interview guides were applied. In an iterative process, two researchers coded the data and derived inductive categories from the text material. In a final pass, two researchers coded the interviews independently using the elaborate category system. Conflicts in coding were discussed among L.L., F.N., and F.K. until a consensualized version for all analyses was completed. All of the coding and analyzing processes were carried out with the aid of the MAXQDA 18 software (VERBI Software, Berlin, Germany) [50]. The interviews were conducted and analyzed in German. In order to make the results available to an international audience, two researchers translated the quotes independently into English. The names of the interviewees were pseudonymized. The thematic qualitative text analysis focused on categories relevant to the research questions, which could be grouped into five main themes (see Figure 1).

## 3. Results

The results from the interviews are presented here for the five main themes (see Figure 1), each of which is discussed below from the perspectives of both the pregnant women and the HCPs. After both perspectives are presented in detail, they are each contrasted in a summary figure at the end of every section (see Figure 2, Figure 3, Figure 4, Figure 5 and Figure 6).

### 3.1. Perspectives on Motivation, Acceptance, and Satisfaction Regarding Lifestyle Counseling in Prenatal Care

#### 3.1.1. Pregnant Women’s Perspective

The women were interested in the intervention, mainly because they expected to receive more extensive counseling for themselves and their babies. Several of the women stated that they believed obesity to be a socially important issue, and that they would like to help to improve care for pregnant women and infants. The first-time mothers were especially interested in receiving more detailed counseling sessions. They often felt uncertain about various issues and were pleased to be given the opportunity to receive extended counseling sessions with HCPs. Some of the women who had already given birth also reported that they were often overstrained, especially during the first pregnancy.
*“because when you don’t have a clue at all and you’re at the beginning and..: Hm, yes, what am I allowed to do now, what should I do, what can I NOT do, what would be better for me? At the beginning, you are a bit overwhelmed when you get your first [baby]”*(Christine, paragraph 67)

The women who had already had children felt that their first pregnancies had already provided them with all of the information they needed. They stated several times that they felt less need to talk. In addition to this, due to their already busy childcare schedules, they had less time to implement the recommendations on lifestyle changes.

The pregnant women were of the opinion that the opportunity for lifestyle counseling should be available as part of routine care, but women should be able to decide for themselves whether and with whom they would like to address the topics, depending on their needs.

Pregnancy is rated as a good time for lifestyle counseling because it is a time when women report taking greater care of themselves. During check-up visits, almost all of the women wanted to discuss what they were allowed to do and what they should avoid. For example, they expected instructions on what foods or sports they should avoid during pregnancy.

The women were mainly satisfied with their participation in the intervention, as it gave them more time to spend with HCPs.
*“I am very pleased. In particular, the additional counseling from the gynecologist was of the main reasons why I participated in this intervention”*(Kerstin, paragraph 86)

Nevertheless, some of the women reported that they already knew everything the HCPs had told them during their counseling sessions. Some of the interviewees pointed out that the counseling should always be adapted to each woman’s individual needs, and that maintaining a healthy lifestyle was already important to them before they became pregnant.

Some of the participants found it difficult to assess whether they had changed any aspects of their lifestyle as a result of the counseling sessions. Nevertheless, they noted that a recommendation from a physician had more impact than when an attempted change was driven by self-commitment alone. For example, one participant identified her unhealthy lifestyle patterns, and now wants to pay more attention to them. She felt that a face-to-face conversation strengthened her focus more than simply reading up on recommendations would. Several of the women reported that jointly agreed goals helped them and provided motivation. They also considered it beneficial to discuss the progress of reaching their goals with their gynecologists.
*“I have to say I really like that because that gives you a little bit of an extra motivation, because every time when checking the app after visiting the doctor, there is a summary of what we talked about and what we agreed upon. That is an additional reminder and then you simply want to accomplish that [goal].”*(Kerstin, paragraph 26)

The pregnant women wanted their counseling goals to realistically fit their daily lives and be easy to implement. Only one of the participants reported that the sessions failed to motivate her at all, and that she already knew everything she was told prior to participating in the intervention.

In summary, minor changes, such as participants eating more fruit or getting up to exercise more, were attributed to counseling. Additionally, some of the women were repeatedly encouraged to exercise by their HCP, even though they had concerns at first.

#### 3.1.2. Healthcare Providers’ Perspective

All of the HCPs interviewed said that their patients generally responded positively to the offer of the intervention. In particular, they reported that the women who were going through their first pregnancies tended to be anxious, and were, therefore, grateful for receiving additional support. Furthermore, some of the women had weight problems during previous pregnancies, and therefore appreciated the counseling sessions.

The HCPs came away with the impression that most of the women were already very well informed prior to the intervention. They often needed reassurance that they were doing things right. When asked, some of the women would also always say that they were doing just fine and did not need the lifestyle advice.

All of the HCPs who were interviewed considered taking the time to provide additional counseling on lifestyle issues to be very worthwhile. They emphasized their intrinsic interest in participating, and noted that they had already dealt with the topics before. Some of the medical assistants stated that they had realized that additional counseling would be beneficial as a result of their own pregnancies. In addition to this, all of the HCPs agreed that there was a need for intervention with regard to overweightness and obesity issues.

Some of the HCPs felt that the counseling had helped the participants. In some cases, awareness was raised regarding the need for change. Sometimes, the help was nothing more than small tips for everyday routines that the patients had not come up with on their own. The HCPs also reported that the joint goal-setting process motivated their patients to give things a try. Some of the HCPs came away with the impression that the women preferred to have their hands held and be given a guideline.

According to one gynecologist, pregnant women are confronted by so many major changes in their life circumstances during pregnancy that they are not able to fundamentally change their diet and exercise if they have not already been eating/exercising adequately. Likewise, this gynecologist believed that women who were already overweight would fail to change their dietary habits, and said that the counseling intervention would thus be unable to help them.
*“I think that during pregnancy, women are confronted with so many things, so many changes in life, that it is DIFFICULT for them to put everything into action, to have adequate physical activity, a healthy diet, when they didn’t even manage to do that before. And that’s what I’ve said right from the start: Those who do that ANYWAY, do not need the program, whereas those who weren’t doing it before pregnancy, definitely won’t manage it during pregnancy”*(gynecologist 5, paragraph 66)

In addition to this, some of the HCPs believed that there were always some women who thought that they already knew everything. This particularly applied to women in their second or third pregnancy. Likewise, there were certain women who were described as resistant to counseling and who did not value additional counseling. Some of the HCPs noted that these were often overweight women who were unwilling to talk about their lifestyle.

One gynecologist had the impression that the counseling was particularly well received by women who were well-educated and physically active, and thus did not really need it. In contrast, another gynecologist explained that he sometimes had to phrase the information somewhat differently depending on the patient’s socioeconomic status, though he would not necessarily say that the better-off knew a lot more. In his opinion, the counseling sessions always needed to be tailored to the patients’ needs and background. In spite of this, some of the HCPs observed an information leak for women with little formal education.

There was consensus that an established relationship of trust between the woman and the HCP, e.g., due to treatment and consultation during previous pregnancies, improved the readiness of the women to accept the counseling.

#### 3.1.3. Summary and Comparison of Perspectives

A summary of the findings and comparison of the perspectives on the motivation, acceptance and satisfaction regarding the lifestyle counselling in prenatal care is given in Figure 2.

**Figure 2 ijerph-19-06122-f002:**
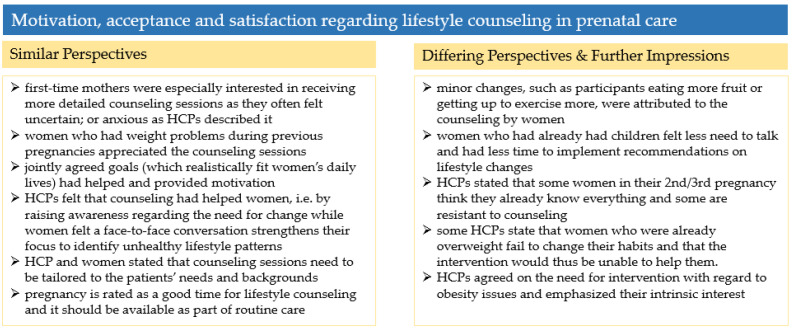
Summary of the results in Section 3.1.

### 3.2. Perspectives on Lifestyle during Pregnancy and Topic-Specific Needs for Counseling

#### 3.2.1. Pregnant Women’s Perspective

All of the women reported that they took more care of themselves during pregnancy. Nearly all of the participants used various pregnancy apps, online searches, and books to obtain information on lifestyle topics. The unborn child motivated them to adopt a healthy lifestyle.
*“[…]and I think, for the good of the child, I think every mom would like to contribute something[…]”*(Elli, paragraph 97)
*“Hm, how can I put it best? It’s about doing my bit to ensure the development of our children”*(Frida, paragraph 63)

Nutrition during pregnancy was considered a very important topic, and advice on it was desired by almost all of the participants. Some of the women expected to be educated on foods that were “forbidden” foods during pregnancy, and to receive a list of rules from HCPs.
*“Yes, so that she [the gynecologist] simply explains, what I can eat, and what’s good for me and what’s not.”*(Christine, paragraph 87)

Some of the participants exercised regularly, but their fitness declined during the course of their pregnancy. The participants were unsure of what activities they were still allowed to do.

During the counseling sessions, nutrition was the most frequently chosen discussion topic. One participant reported that she had more in-depth counseling sessions on nutrition due to her gestational diabetes. Another participant needed specific advice because she wanted to maintain her vegetarian diet. In addition to nutrition, the integration of physical activity into the women’s day-to-day routines was also discussed, as well as sufficient water intake. Smoking and alcohol were not discussed in depth because they were of no concern to any of the women who were interviewed.

One interviewee stated that she knew enough about the topics herself and therefore did not want to waste time receiving counseling on lifestyle issues. She believed that people thought enough about healthy lifestyle choices without the need for further advice. She had gained more weight than she wanted, and considered this to be due to a lack of physical activity.

The women reported that they would also like something to take home after the counseling session, such as an information brochure on the lifestyle topics they had discussed. The participants reported that their minds were often very busy during the counseling sessions, and that it would be great to be able to remind themselves of the conversation using written information the next day.

The predefined topics corresponded to the participants’ expectations. Most of the women felt that, in addition to these topics, they could also address any other issue as necessary. One participant said she would also be open to home visits for counseling sessions on breastfeeding.

#### 3.2.2. Healthcare Providers’ Perspective

The HCPs believed there was a tremendous need for lifestyle counseling, since they provide care to many overweight women. One gynecologist said that the needs of pregnant women varied greatly depending on their initial weight and level of education. One gynecologist said that many women had no idea what healthy food was, and that they stopped exercising the moment they discovered they were pregnant.
*”because they simply have no idea at all what is healthy food and what is not. They put themselves to bed: I’m not moving (laughs slightly), that could harm the child (laughs slightly). That’s really blatant”*(Gynecologist 3, paragraph 8)

One medical assistant came away with the impression that the women were mostly asking for confirmation on whether they were eating enough and whether their diets were healthy enough.
*“I would say that nutrition [is the most important topic for women]. Many are uncertain about this. Am I now eating sufficiently, am I now eating HEALTHY enough? So I always have this feeling, yes.”*(Medical Assistant 3, paragraph 44)

The HCPs confirmed that nutrition was the most popular counseling topic, followed by physical activity. They also stated that nutrition was usually particularly important to women during their first pregnancy. One gynecologist said that the participants often had problems with gaining weight or drinking water. Some physicians stated that alcohol and nicotine-related issues were a problem. Smokers often do not manage to quit completely, while alcohol consumption is very taboo and often kept secret. The gynecologists stated that many problems, such as substance abuse disorders, cannot be addressed in regular preventive care, and said that some women also needed psychological support.

One gynecologist reported that it was difficult for the participants to decide which lifestyle topic they wanted to discuss while still in the early phase of pregnancy. During this phase, worries and fears regarding the progress of the pregnancy are still highly prominent. In addition to this, the early stages of pregnancy involve a large number of medical tests and require the women in question to handle a multitude of information.
*“the pregnant woman COMES to the determination of the pregnancy, then one determines the pregnancy and then she is OVERCOME first with completely many information. Right? And there are really MANY, MANY, MANY things, so she must first come to terms with the fact that she is pregnant at all, is happy or not happy, is afraid whether the pregnancy will go well or not—you don’t know at the beginning of the pregnancy. Then (clears throat) is the explanation, okay, now maternity care starts. What does prenatal care mean, what do all the examinations that are done in prenatal care mean?”*(Gynecologist 4, paragraph 12)

As a result, they cannot remember everything. Due to this, some of their patients expressed disappointment that they did not receive any written information after the counseling sessions. They also noted that pregnant women needed to adjust to their new life circumstances, and did not consider lifestyle issues a priority for this reason.

#### 3.2.3. Summary and Comparison of Perspectives

In Figure 3, the results on lifestyle during pregnancy and topic-specific needs for counselling are summarized and the perspectives on this main theme are compared. 

**Figure 3 ijerph-19-06122-f003:**
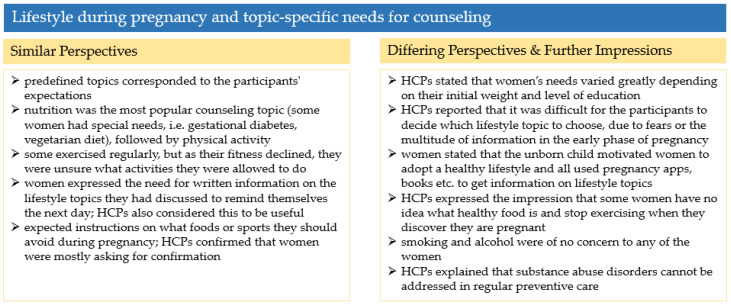
Summary of the results in Section 3.2.

### 3.3. Perspectives on Gestational Weight Gain and Needs for Counseling

#### 3.3.1. Pregnant Women’s Perspective

For the women who participated in the study, weight gain was seen as a normal part of being pregnant. The participants gave the impression that they were not particularly concerned about weight gain, and did not think they could do anything about it anyway. None of the participants associated weight gain with consequences for their own health or that of their child.
*“I make sure that it’s not so MUCH [weight], but I/Now if it’s 15, 20 kilos, then that’s just how it is [...] So it’s just pregnancy (laughs lightly), so then you gain weight, right?”*(Christine, paragraph 54)
*“Actually, it [weight gain] does not matter so much now. What is certain is that you gain weight. I am not exactly the skinniest of the participants. But I’m not worrying about it right now.”*(Elli, paragraph 26)
*“Well, I mean, you can’t really influence it [weight gain] much, or you shouldn’t really influence it much, by saying: Oh dear, I’m putting on far too much weight, I want to cut back. So I wouldn’t do that, also with regard to the health of the child, that the child would then, I don’t know, suffer any disadvantages in its development.”*(Frida, paragraph 28)

Some of the interviewees seemed to be of the impression that they did not need to be counseled regarding weight gain, even if they had already gained a lot of weight or started their pregnancy at a high initial weight. One of the participants explicitly stated that she had gained very little weight, and, therefore, did not need to talk about weight. Some of the women reported that their weight was not discussed with a gynecologist or midwife at all. Others reported that sometimes, after weighing, they had been told that their weight gain was within limits, but that there was no further conversation on the topic afterwards.

Only one of the interviewees reported that her gynecologist had discussed and analyzed her weight gain with her. At the beginning of the pregnancy, she was afraid of gaining the same amount of weight as she had during her previous pregnancy. As a result, she was appreciative of the helpful advice on nutrition during the consultation.

One participant explained that she had gained a lot of weight, but said that she did not need to talk about it because she knew herself what had caused the gain. Her gynecologist advised her to write down her daily meals in spite of this, and she now reports that she is in better control of her weight.

In summary, it seems that none of the women were aware of weight gain recommendations or the risks associated with excessive weight gain.

#### 3.3.2. Healthcare Providers’ Perspective

The HCPs possessed differing views on the relevance of gestational weight gain. There were both midwives and gynecologists in the study who believed that it was not their job to talk about weight, and stated that they had many other important priorities.
*“So I think as long as she feels good and does not have any side effects, so if blood pressure is okay, it’s not important for me whether she gains 16 or 18 or 20 kg.”*(Gynecologist 1, paragraph 56)

Some midwives even said that they did not want to address weight gain because it felt uncomfortable.
*“You just have to be a little bit careful, and when I don’t see the women during the course of the pregnancy, and only at these counseling sessions, I’m just a little bit more cautious about bringing up the subject of weight if it would be extreme in any way.”*(Midwife 3, paragraph 24)

Moreover, some of the HCPs reported that they had had difficulty communicating recommendations regarding gestational weight gain to overweight women. One gynecologist believed that to do so would be in conflict with the MI technique, as consultants should not give instructions when using MI. In contrast, one medical assistant said that MI techniques were helpful because they provided a means of approaching the topic of weight gently and sensitively.

On the other hand, there were also gynecologists who said that they always addressed weight, and see regular weighing during check-ups in particular as an opportunity to repeatedly raise awareness. Their impression was that women were more sensitized to the issue of their weight when it was discussed frequently. In their opinion, a combination of regular weighing and information dissemination had the potential to change lifestyles. They, therefore, believed that pregnancy and the close accompanying monitoring can be particularly beneficial in this regard.
*“So, of course, all you need is information, and also of course this/We weigh them every four weeks. They’ll never have that again in their lives, right? So then they’re like: (changes voice pitch) Oh, my God, I don’t want to be asked about it again at the gynecologist.”*(Gynecologist 3, paragraph 56)

Another gynecologist said that his patients know how strict he is with regard to weight gain. Even outside of pregnancy, he discusses options with obese women or refers them to colleagues.
*“and then pregnancy starts, and I say “Yes, you know, weight development, how high it SHOULD be” and then you can see how it develops and that’s good […] So it seems to help if you keep pointing it out.”*(Gynecologist 2, paragraph 26)

Another of the gynecologists said that, although she tries to address weight frequently, women have a very different focus and want to know if their child is healthy. Often, her patients are more concerned when they are perceived to not be gaining very much weight.
*“The focus is on the child. After that, whether they’ve gained a lot of weight or not is only a minor concern. That’s something that doesn’t really interest them deeply. Funnily enough, it’s more the NOT gaining weight. The significant weight gain shocks them rather less (laughs).”*(Gynecologist 4, paragraph 26)

One of the gynecologists noted that, for obese women, body weight is without a doubt an issue before pregnancy and that it should ideally have been talked about beforehand. In contrast, another of the gynecologists explained that she would only discuss lifestyle issues in the context of prenatal care, because, in such scenarios, they also have a direct impact on the health of the child. Outside of pregnancy, she sees no obligation to address the issue, and considers it the responsibility of a general practitioner.

One of the gynecologists was convinced that pregnant women are concerned about their weight because they are constantly being asked about their appearance. Nevertheless, most of her patients were unaware of the recommendation. Practically all of the HCPs observed that the women were not familiar with the recommendations for adequate weight gain during pregnancy.

#### 3.3.3. Summary and Comparison of Perspectives

Figure 4 summarizes the findings on gestational weight gain and needs for counselling and compares the perspectives of pregnant women and HCPs.

**Figure 4 ijerph-19-06122-f004:**
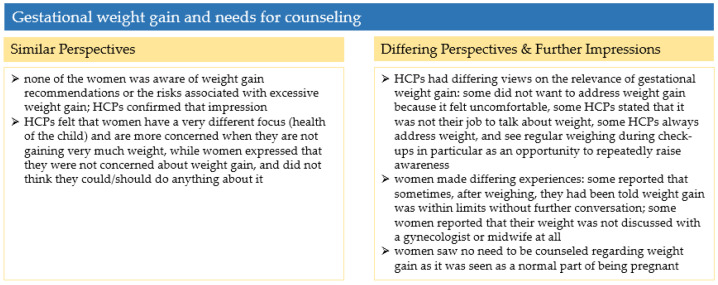
Summary of the results in Section 3.3.

### 3.4. Perspectives on the Appropriateness and Feasibility of Embedding Counseling Sessions into Routine Prenatal Check-Up Visits

#### 3.4.1. Pregnant Women’s Perspective

In all cases, the women appreciated the fact that the counseling sessions were carried out as part of their routine prenatal care.
*“Yes, I think so. Because where else can you go/I think it makes sense when you’re at the gynecologist’s that you also talk about such topics.”*(Elli, paragraph 58)

The majority of the participants were opposed to additional appointments outside of their regular check-up visits. The pregnant woman also said they would also only consult other healthcare experts outside of their routine prenatal care setting if problems arose. For example, one participant said she could see herself contacting a lactation consultant if her breastfeeding was not going well.

The women provided highly differing descriptions of their counseling sessions. Some women felt that a lot of time was given to them. Others complained that there was little time for a conversation, and that things were rather hectic. One woman said that she stopped asking questions because everyone in the practice was so stressed. Some women reported that, despite being enrolled in the trial, they had not yet received counseling, nor had their lifestyle issues been addressed. However, the women also found it difficult to distinguish their standard care from the intervention.

Most of the interviewed women received lifestyle counseling at their gynecological practices. In half of the sample, there was no involvement at all from medical assistants in the intervention components. In some cases, they assisted with documentation or with preparing topics for the counseling sessions. For example, some medical assistants attempted to identify the topic the patients wanted to discuss. Two women reported that they had received counseling from medical assistants. Only one of the women who were interviewed received counseling from a trained midwife. The other participants reported that they only saw their midwives at a later stage of their pregnancy.

About half of the women who received counseling sessions chose the counseling topic themselves. The topics for the other half of the sample were predetermined by the respective HCP. From the interviews with the women, it appears that the HCPs often asked questions regarding their behaviors, then offered recommendations in response.
*“For example, when it comes to eating behavior, she first asks me what I like to eat or what I eat in general, i.e., whether I eat healthily or not, or when it comes to drinking, what I drink all day, how much I drink and (clears throat) I answer all the questions. Then, if she has any other information that doesn’t match my questions, then she informs me about it.”*(Doris, paragraph 12)

#### 3.4.2. Healthcare Providers’ Perspective

All of the HCPs considered prenatal care to be an appropriate setting for preventive counseling. The gynecologists stressed that a gynecological practice is a good setting for preventive counseling because they usually already have a long-standing relationship with their patients and see them regularly. Emphasis was also placed on the fact that prenatal check-up visits at a gynecological practice present a reliable opportunity to speak to women about their health, since all women attend these services. Medical assistants can usually schedule appointments in order to tie the consultations to regular check-up visits.

The gynecologists did not take patient accessibility via midwives as a given, as many women are not in contact with midwives during their pregnancy; in fact, some have no contact with midwives at all. The gynecologists also pointed out that a medical practice provides a safe space where these conversations can take place uninterrupted. The gynecologists usually incorporated their consultations into the regular check-up visits. Some took 5–10 min for the consultation, and others between 15 and 20 min.

On the other hand, all of the gynecologists reported a lack of time due to many other issues relating to regular screening during check-up visits. One gynecologist stressed that gynecologists are mainly responsible for curative matters, and that preventive medicine is not something they generally deal with.
*“Preventive medicine in general just basically isn’t something we do, we are basically there for curative issues. But then that’s a contradiction in itself, because there is no curative activity for us to carry out in maternity care. So we definitely need to talk about the extent to which such a practice procedure really offers room for it. But, yes, on the other hand, this is again contrary to the relationship work that one does as a caring doctor.”*(Gynecologist 4, paragraph 64)

One gynecologist explained that she needed to educate the women on numerous topics, and suggested that midwives should be made more aware of prevention topics. However, she also pointed out that midwives all have different levels of training. Despite this, the gynecologists stated that breastfeeding was a topic traditionally discussed in midwifery.

Several of the HCPs did not apply the conversational MI technique, deciding instead to stick to their usual conversational approach. One physician stated that he did not consider the technique applicable at all. One of the gynecologists considered MI inappropriate for topics such as breastfeeding.

All of the midwives stressed that they had always provided lifestyle counseling and saw themselves as suitable counselors, since they also assisted families after the birth. Nutrition and breastfeeding have always been topics on which midwives have provided detailed counseling.

Contrary to the study protocol, all of the midwives reported that they always made additional appointments for lifestyle counseling as part of the intervention, as they did not normally see their patients until shortly before birth. The midwives visited the women in their homes and spent about 20 min on counseling. They felt that going to the woman’s home specifically for this purpose gave the consultation special relevance. The midwives also highlighted a number of other advantages to providing counseling in the home environment—there were no interruptions, they were able to take more time for the conversation, and they also gained an insight into the women’s lifestyles in their homes. Nevertheless, they noted that the visits were time-consuming and not very profitable. In terms of scheduling, they said that the facts that they do not have practice offices and that it is difficult to coordinate on-site home visits were problematic. One of the midwives said that they would like a predefined guideline on how to incorporate the counseling sessions into her workflow. On the other hand, another of the midwives expressed concern that gynecologists’ offices are too overburdened, and said that midwives can be more flexible and provide longer counseling sessions on an individual basis.

One gynecologist pointed out that the quality of counseling varied greatly among all colleagues. In addition, he emphasized that, in the gynecological practice, they can only cover the tip of the iceberg and highlight topics. He refers obese women to nutritional counseling and draws their attention to the services offered by health insurance companies.

Another of the gynecologists expressed concern that dedicated and well-educated women would follow the recommendation to see a nutritionist when they were actually the group that least needed to do so.
*“So I think that it [the gynecological practice] is the right place, because they will definitely be there. [...] So if we now say that they should all go to a nutrition consultation, then I’ll tell you: All the working women won’t go, they’re happy when they’ve managed to get the appointment here, ok? All those who more or less let everything slide anyway, i.e., the unmotivated ones, they will NOT go either. Then the women you have in the nutritional counseling are the ones who actually don’t really need it, because they’re already quite good anyway.”*(Gynecologist 3, paragraph 124)

Some of the HCPs stressed that the program was unable to reach the women who needed to be addressed most urgently. All of the HCPs agreed that there was an urgent need to find a way of conducting good counseling sessions with non-German-speaking women. In addition to this, they said that all of the information materials needed be translated as standard.

Another of the gynecologists reported that most of her patients had a huge need for counseling on childbirth, and many fears and concerns that needed to be discussed. She said that sometimes there was more focus on this than on lifestyle issues. This gynecologist suggested using the counseling time to discuss all of the patient’s fears first, otherwise, the women would not be able to concentrate on lifestyle issues.

One of the gynecologists said that she would like to see general changes in the health care system, and that it was not cost-effective for her to conduct in-depth consultations with her patients. She claimed that HCPs needed more time and adequate compensation. Likewise, the midwives said that they would like to be reimbursed for the consultation in a manner similar to a postpartum visit. In addition to this, it was agreed that regular training should be provided. Some of the gynecologists also suggested that medical assistants should be more closely involved in the consultation process. The medical assistants echoed this preference.
*“I have an additional qualification as a nutrition consultant and […] I find it especially interesting in pregnancy and that was my motivation for me. […] I would like to do more personally, but I’m kind of not allowed to. So I think that’s a bit of a shame”*(Medical Assistant 1, paragraph 54; 92)

#### 3.4.3. Summary and Comparison of Perspectives

A summary of the findings and comparison of the perspectives on the appropriateness and feasibility of embedding counselling sessions in routine prenatal check-up visits is given in Figure 5.

**Figure 5 ijerph-19-06122-f005:**
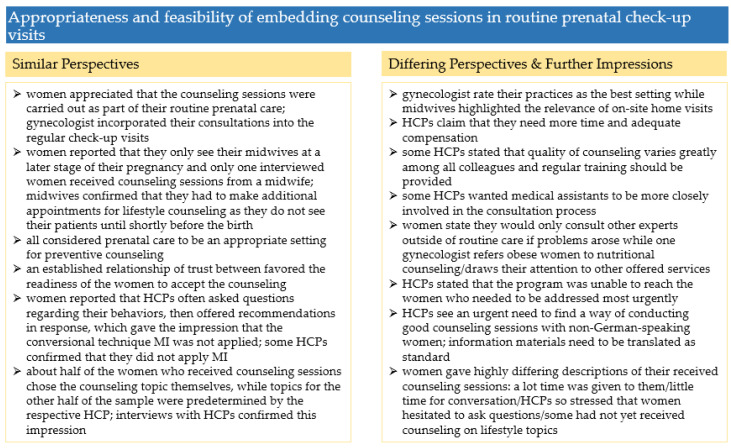
Summary of the results in Section 3.4.

### 3.5. Perspectives on Inter-Professional Cooperation and Receiving Counseling from Different Healthcare Providers

#### 3.5.1. Pregnant Women’s Perspective

Several of the women liked the idea of receiving lifestyle counseling from multiple HCPs. They felt that the more often they heard the key messages, the better. In addition to this, they believed that it would be a good idea for all of the professions involved to consult on lifestyle topics, as they hoped that this would give them a more comprehensive picture and the opportunity to explore different perspectives. In contrast, one of the women, who had already given birth to several children, said that she would have liked to choose who her counseling session was with, and did not want to have to discuss the topics with everyone.
*“I am not sure whether I would be annoyed by this, when visiting all three providers, […] I would say (sighs) one time would be enough. So I think it would be good if you could choose, so everyone offers it and you can decide who you trust the most. But hearing it from everyone, I think that is too much.”*(Helga, paragraph 36)

Some of the women said that they only saw their midwives shortly before/after giving birth, or only for a birth preparation class. As a result, they had no counseling sessions with their midwives. In some cases, the women already knew their midwives from previous pregnancies, and said that there was no need to see them early.

The women described a relationship of trust with their HCP as being particularly crucial for counseling. Which HCPs were trusted varied greatly from one woman to the next. Some participants reported that they already had a relationship of trust that had been established during a previous pregnancy. One of the participants felt that the gynecologist was the best person to provide the counseling, but said that she would still like the midwife to be more involved. One participant specifically said that she would prefer to confer with her gynecologist because, unlike the midwife, the gynecologist was someone who would continue to provide her with medical assistance for many years to come.

Some of the participants experienced a close relationship of trust with their midwives, and said that they would particularly like to receive advice from their midwives on breastfeeding. One participant said she would like to discuss all of the topics with her midwife, because she sees the midwife both during and after birth. Another of the women also placed considerable trust in her midwife, as she felt it was safe to assume that the midwife would have a particular interest in ensuring that the birth was free of complications. One of the participants reported that her midwife was available to her at all times and always responded promptly. In contrast, she hardly felt comfortable asking any questions at all at her gynecological practice.

The pregnant women expressed uncertainty regarding the relationship between gynecologists and midwives. Some of the women explicitly requested that the HCPs not contradict each other in counseling. The women were under the impression that midwives and gynecologists do not exchange information with one another and do not have access to the same data. In addition to this, the women assumed that HCPs do not maintain any contact with each other. Some of the participants were highly dissatisfied with the lack of collaboration, saying that there seemed to be a lack of mutual acceptance and respect.

The participants felt torn between their gynecologists their and midwives. They felt that some gynecologists seemed to believe that a midwife was not needed, while the midwife had offered to take over the preventive care.
*“My midwife offered to do the usual prenatal care, just like the doctor would do it. That would be my choice, whether seeing the doctor or seeing her. They are both from this village, and she made the remark that my gynecologist is not convinced about letting the midwife do that and said I don’t need a midwife anyway, and that’s why I am thinking there is no cooperation between them.”*(Frida, paragraph 46)
*“Yes, I would say it [cooperation] is quite bad. I have a midwife who I am visiting for every second prenatal care appointment, because I want to give birth in a birthing center. And it seems like my gynecologist does not accept that. Every time I visit her she keeps saying to me that I should make the next appointment for in about two weeks, and I am not familiar with the legal situation of what is my right, and every time I see my midwife she keeps saying that my gynecologist did too much, and she wasn’t allowed to do that, because it was agreed that my midwife would do that. That is a difficult situation for me.”*(Brigitte, paragraph 49–50)

In addition to this, the midwives and the gynecologists offered differing advice on a number of topics. One participant reported a discrepancy between the information she had received from her gynecologist and that from her midwife. For example, the midwife might have recommended something, then the gynecologist would state that the proposed action would not be of any help, and, as a result, the participant would not know what to do. At the same time, some of the women described midwives as peculiar, and said that they were thus hesitant to follow their advice. In this context, the women described their physicians as the authority.
*“Midwives are usually kind of a bit, let’s call it ‘special.’ Every one of them has her direction where she’s heading and she is super convinced of that, but I am not sure if they are able to judge objectively. Every one of them has her own, let’s call it ‘style.’ So I would maybe rather lean towards the doctors.”*(Frida, paragraph 56)

One participant said that she was more likely to listen to or act on something a doctor might say than a midwife. The women were not generally referred to other health care experts. Unless there was any particular need, they might not think of visiting other experts. Two of the participants were diagnosed with gestational diabetes, and were, therefore, referred to a diabetologist.

#### 3.5.2. Healthcare Providers’ Perspective

All of the HCPs said that there was a need to engage more with their colleagues regarding counseling on lifestyle topics. All of the HCPs also reported that the intervention had not led to any changes with regard to collaboration.

One of the gynecologists has always worked hand-in-hand with midwives in her practice; three midwives rent offices in her practice and the collaboration works very well. The gynecologist carries out the preventive care first, then the women usually go to see the midwife afterwards. The gynecologist in question strongly supports this approach. In her opinion, gynecologists and midwives have different areas of expertise, and, therefore, complement each other well. Nonetheless, she expressed concern that this is not the way things are done in most practices. She believed that legislation has hindered collaboration between midwives and gynecologists, and said that this was bad for all of the parties involved.
*“It has also been hindered by the legislation. […] This is not good for the pregnant women, for pregnancy counseling, for the midwife, and not for the doctors either, right? Nobody knows what that was all about. But (...) midwives can do different things to me. And I can do different things to the midwife. And of course I do my regular prenatal care, that’s obvious, that’s also obligatory, that’s how it should be, that’s what the women want. But they come HERE because they read on the Internet that I work with midwives, right? And then that’s exactly how it is: they have their own consultation hours, and then the patients can just go there additionally.”*(Gynecologist 3, paragraph 156)

The other participating gynecologists reported that they had no contact with midwives. One gynecologist expressed regret at this, as she believes that messages are received better when they come from different HCPs. She would be open to gynecologists and midwives sharing prenatal care in a better way. For example, gynecologists could focus on more of the technically related matters and midwives could conduct more of the preventive work.
*“in this room, the pregnant women are perhaps more receptive […], because they are more focused on getting this information, and if one were to speak the same language and the pregnant women knew, okay, my midwife says this, and my doctor says the same thing, so in that imaginary scenario, okay, it’s my job as a doctor to somehow record the technical points and perhaps then consult with the midwife. Maybe I would advise her to pay a little more attention with one patient, or discuss what could be done with another one, but then I would leave the intervention itself to the midwife.”*(Gynecologist 4, paragraph 72)

The remaining gynecologists expressed little interest in working with midwives. One gynecologist explained this by saying that they did not have time to network. Another of the doctors had had bad experiences in the past, and said that midwives had made questionable recommendations he did not agree with. Nevertheless, he recognized that midwives perform an important job and can offer women a closer level of care than a gynecological practice is often able to. Due to the shortage of midwives, the gynecologist in question said that he already advises all newly pregnant women to seek midwifery care as soon as possible.

One of the gynecologists said that he was not interested in networking and discussion because, firstly, he had no further use for other people’s information, and secondly, he did not want to interfere with anyone else.

The medical assistants reported that discussion and collaboration in a large practice is difficult because it is not clear which midwife is in charge of which pregnant woman.

One of the midwives described the nature of the communication between physicians and midwives as old-fashioned: the midwife approaches the physician, but not vice versa.
*“We midwives have been thinking about this for a long time, but it’s hard to get the doctors to do it. So we go to them, but they don’t come to us (laughs slightly) […]. I think that’s just an old-fashioned attitude to collaboration in general, which is certainly almost historically conditioned.”*(Midwife 2, paragraph 122–124)

One midwife suggested that the lack of discussion was due to tight schedules and the overburdening of both physicians and midwives. In addition to this, competitive thinking could also play a major role. One midwife observed that women were more likely to follow advice from gynecologists than that from midwives.

The midwives in particular indicated that they would like to see an improvement in their collaboration with gynecologists. They all considered joint training to be beneficial, and emphasized the importance of understanding the respective skill sets of each professional group and the way in which each one consults. They saw knowing one another’s faces as important in facilitating the exchange of patient information and further referrals. In addition to this, they advocated for a more holistic approach to counseling during pregnancy.

#### 3.5.3. Summary and Comparison of Perspectives

The results and perspectives on inter-professional cooperation and receiving counseling from different healthcare providers are summarized and compared in Figure 6. 

**Figure 6 ijerph-19-06122-f006:**
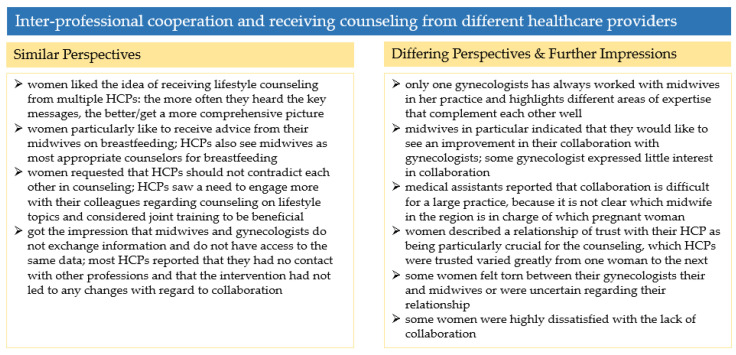
Summary of the results in Section 3.5.

## 4. Discussion

The results of this study are valuable for tailoring preventive measures in prenatal care according to the needs and expectations of pregnant women and their HCPs. The findings illustrate the similarities and differences in the expectations and experiences of women and HCPs with regard to the preventive counseling in pregnancy provided in the GeMuKi intervention. This demonstrates the importance of including both patients’ and HCPs’ perspectives when planning and designing implementation.

The pregnant women expressed a need to talk about lifestyle issues, mainly in terms of nutrition and physical activity. The first-time mothers in particular felt a great need for counseling and welcomed the extra time with HCPs. This was reflected by the HCPs in their daily practice as well. Furthermore, the HCPs pointed out a tremendous need for lifestyle counseling, since they provided care to many overweight women.

All of the pregnant women who participated in the study stated that they wanted to strive for a healthy lifestyle in order to benefit themselves and their child. This behavior was not questioned and could represent a form of social desirability. Atkinson et al. (2016) found that women whose pregnancies were not characterized by a sense of vulnerability or anxiety made lifestyle decisions based upon a “combination of automatic judgements, physical sensations, and perceptions of what is normal or ‘good’ for pregnancy” [18]. Furthermore, Rockliffe et al. (2021) found that women wanted to adopt to the role of the ‘good mother’ by making healthy lifestyle changes, but, at the same time, a lack of understanding with regard to health consequences and low risk perception represented barriers to change [51].

The interviews emphasized that perspectives on gestational weight gain varied widely. Pregnant women assumed that they could not influence gestational weight gain and did not link it to the health of the child. Although the HCPs described the women as well informed, the HCPs believed that the women were not aware of recommendations for weight gain during pregnancy. Despite this, HCPs differed in how and whether they addressed weight gain, if they did so at all, and what relevance they attached to it. Moreover, some HCPs reported difficulties in communicating gestational weight gain recommendations to overweight women.

This is in line with findings that stated that pregnant women were not aware of the risks associated with gestational weight gain [37,52,53]. Pregnant women often base their behavior regarding diet and physical activity on their social and community environment and their peers’ beliefs [54,55]. While risks, such as smoking during pregnancy, are discussed in these contexts, the risks relating to weight gain are often not known and are not talked about [55]. This further highlights the importance of sharing information on gestational weight gain through HCPs. There is evidence that women who have received information from their gynecologists have a higher level of knowledge with regard to lifestyle-related factors during pregnancy [56]. Liu et al. (2016) showed that weight gain recommendations made by HCPs are an important predictor of actual weight gain [57]. Furthermore, Deputy et al. (2018) found that both inadequate and excessive weight gains were more likely in women who had received no recommendation at all [58]. Research has also indicated that pregnant women assume that weight gain is not a relevant issue if it is never addressed by HCPs [59]. Additionally, findings illustrate a need for accurate advice from HCPs regarding gestational weight gain recommendations [60]. Research is needed on appropriate resources and materials to support HCPs in giving consistent weight gain advice [36].

All of the interviewees agreed that regular check-up visits in prenatal care were a good setting for lifestyle counseling. While the HCPs reported a lack of time due to many other issues related to regular screening, the women appreciated the fact that they did not have to attend additional appointments for lifestyle counseling outside of their normal check-up visits. Embedding additional counseling into routine care was not always feasible for midwives, while it was easy to organize in gynecological practices. While this was not a concern of the interviewed women, some HCPs pointed out that the intervention was unable to reach the women who needed to be addressed most urgently. More research is needed regarding methods to improve outreach to these women and to refer them to experts.

All of the interviewees agreed that joint goal setting and reminders may help pregnant women in making lifestyle changes. Aside from incorporating joint goal setting, the best approach for counseling on lifestyle-related topics remains unclear. The MI technique was not always used and some of the women tended to expect concrete instructions, rather than an open conversation. In contrast, the HCPs stressed that MI techniques had been particularly helpful in enabling them to address difficult and sensitive topics, such as weight. This is in line with other findings, which demonstrated that implementing MI techniques can facilitate openness and create trust, but pose challenges to medical practices due to a lack of time in their daily routine [61,62].

However, it is important to consider that HCPs should be trained in sensitive communication. There is a risk that HCPs who are not trained and not aware of obesity and lifestyle issues may provide discriminatory advice. HCPs, therefore, require additional training to ensure that they do not stigmatize their patients and inadvertently harm the relationship or health outcomes [63,64]. Continuing education on lifestyle counseling could also benefit patients in other stages of life, such as those undergoing hormonal changes during menopause or cancer and cardiovascular disease [32].

The pregnant women described a relationship of trust with their HCP as particularly crucial for counseling. They were dissatisfied with the collaboration between gynecologists and midwives. Conflicts between the professional groups were sometimes acted out at the patients’ expense, resulting in insecurity. The midwives in particular expressed a desire for improved cooperation, while the gynecologists mostly believed that discussion was only needed if complications occurred. Many women do not receive care from a midwife until the last few weeks before birth. Some of the interviewed gynecologists proposed a better division and coordination of consultations so that each profession could focus on their respective field of expertise. Interdisciplinary stakeholders in health care relating to childbirth in Germany have also called for improved collaboration, for example, through joint education and training, and resolution of legal ambiguities [65]. Different authors point to the importance of commitment, interpersonal skills, effective communication, respect, and trust among HCPs for successful collaboration [66,67,68]. More research is needed to examine the deep-rooted reasons for the difficulties in collaboration between gynecologists and midwives in Germany. Van der Lee et al. (2016) described a combination of exploring contemporary inter-professional practice with a historical perspective on inter-professional collaboration as beneficial to understand problems, and to provide guidance for improving collaboration [69]. From this, implications for policy and practice could be derived and could enable practitioners to implement actions for improving collaboration.

### Strengths and Limitations

One strength of the study was the open and explorative character of the interviews. At the beginning, the women were asked to tell the interviewer about their last counseling session with their gynecologist and/or midwife. This led to an open flow of conversation in which the women were able to decide for themselves what to focus on. Another strength was the study’s ability to incorporate inter-professional perspectives, as it allowed gynecologists, midwives, and medical assistants to share their experiences. The fact that different researchers were involved in the iterative analysis process represents another advantage, as it meant that the results were discussed in depth at various stages and according to the text material.

As shown in an evaluation of the recruitment procedures during the GeMuKi trial, intrinsic motivation was one of the major factors that led to HCPs participating in the GeMuKi trial [70]. The HCPs who consented to be interviewed were most likely motivated. It was, therefore, reasonable to assume that they did not represent typical HCPs in terms of implementing the intervention. A larger sample of different healthcare providers would have been beneficial. Unfortunately, it was not possible to recruit more healthcare providers for an interview. The interviews did not provide the information required for a comprehensive evaluation of the use of MI techniques. This would have required recurring observations of the counseling sessions, which was unfortunately not possible in practice.

## 5. Conclusions

Pregnant women and HCPs rated regular check-up visits during pregnancy as a good setting in which to focus on lifestyle topics. In particular, both pregnant women and HCPs reported that the combination of joint goal setting, reminders via push notifications, and feedback sessions helped women to make minor lifestyle changes. Nevertheless, it became apparent that there was a lack of information among pregnant women with regard to the recommendations for adequate gestational weight gain, and that the counseling approaches adopted by HCPs varied greatly. A discussion should be held regarding using sensitive techniques to inform all pregnant women of the risks and consequences of excessive weight gain. In addition to this, strategies should be sought to improve inter-professional collaboration between all of the HCPs involved in regular prenatal care. The results of this study will help to improve health care in pregnancy by taking into account the perspectives of both pregnant women and their HCPs.

## Figures and Tables

**Figure 1 ijerph-19-06122-f001:**
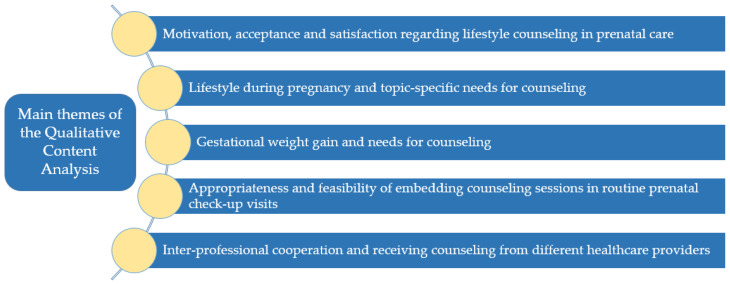
Main themes of the Qualitative Content Analysis.

**Table 1 ijerph-19-06122-t001:** Sample description of pregnant women; mean values (minimum; maximum).

	Participants (*n* = 12)
Interview duration (minutes)	21:16 (15:00; 26:44)
Age (years)	32.5 (30; 37)
Week of pregnancy	32 (28; 36)
BMI before pregnancy	25.64 (21.64; 33.06)
Parity	No children: 50.0% (*n* = 6) One child: 33.3% (*n* = 4) Two or more children: 16.7% (*n* = 2)

**Table 2 ijerph-19-06122-t002:** Sample description of HCPs; mean values (minimum; maximum).

	Gynecologists (*n* = 5)	Assistants (*n* = 5)	Midwives (*n* = 3)
Interview duration (minutes)	40:00 (25:00; 60:00)	17:12 (7:00; 25:00)	28:20 (25:00; 30:00)
Gender	Male: 1/5 Female: 4/5	Male: 0/5 Female: 5/5	Male: 0/3 Female: 3/3
Professional experience (years)	8 (4; 16)	20,67 (5; 32)	22,67 (9; 42)
Office size	9 (3; 16)	9,8 (5; 20)	-
Employment relationship	-	-	Employed: 0/3 Self-employed: 3/3

## Data Availability

The datasets used and analyzed in this study are available from the corresponding author on reasonable request.

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
