# Peer review of "Preventive Counseling in Routine Prenatal Care—A Qualitative Study of Pregnant Women’s Perspectives on a Lifestyle Intervention, Contrasted with the Experiences of Healthcare Providers"

_ijerph, 2022, doi:10.3390/ijerph19106122_

Round 1

Reviewer 1 Report

Dear Authors,

thank you for the possibility to review this fine paper.

Overall, I think the research was well and rigorously conducted. The paper is well written and logically constructed. I only suggest you to spend more words in the discussion section, making more comparisons with previous literature and providing suggestions/tips for future research and clinical practice. In particular, I will deepen the discussion concerning the poor cooperation between gynecologists and midwives, suggesting hypothesis to be tested in future or tips for interventions. Could the leadership style impact the cooperation of healthcare professionals? Could the fostering of a positive ethical climate or social support improve the collaboration in the team? Concerning this, take a look to the following references:

Giovanni Gibiino, Michele Rugo, Marina Maffoni, Anna Giardini, Back to the future: five forgotten lessons for the healthcare managers of today, International Journal for Quality in Health Care, Volume 32, Issue 4, May 2020, Pages 275–277, https://doi.org/10.1093/intqhc/mzaa021

I wish you all the best with your paper.

Reviewer 2 Report

Dear authors,
I congratulate you for the article "Preventive Counseling in Routine Prenatal Care. A Qualitative Study of Pregnant Women's Perspectives on a Lifestyle Intervention Contrasted with the Experiences of Health Providers."
The research article meets the following criteria:

1. The study presents the results of original research.
2. The reported results have not been published elsewhere.
3. Experiments, statistics, and other analyzes are performed to a high technical standard and described in sufficient detail.
4. Conclusions are adequately presented and supported by data.
5. The article is presented in an intelligible manner and is written in standard English.
6. The research complies with all applicable standards for the ethics of experimentation and the integrity of the research.
7. The article adheres to appropriate reporting guidelines and community standards for data availability.

Reviewer 3 Report

Dear authors,

I think that this work is of interest and is well executed.

As for strength and weakness, the only thing that I could suggest for future work is a greater sample for healthcare providers, due to the different functions of the different professionals.
Surely, it is a very delicate topic, especially due to the particular moment in which women have to care directly not only for themselves but also for a little human. This is a huge responsibility which is often dismissed by healthcare professionals, since they consider it normal, but for some women it may be a great burden. This is why I feel that this work has the merit to dive deeper into this subject.

Please check and proofread the paper, since there are some typos and phrases that need to be fixed.

Best regards

Reviewer 4 Report

Thank you for the opportunity of reviewing this manuscript entitled: “Preventive counseling in routine prenatal care. A qualitative  study of pregnant women's perspectives on a lifestyle intervention, contrasted with the experiences of healthcare providers”.

Line 39-40. I think that authors should recommend preventive interventions, even before pregnancy as some authors declared.

The paper focuses on gestational weight gain and maternal and infant health, but is more general, about lifestyle during pregnancy. This point should be taken into account.

References do not follow the structure required by the journal.

The text needs to correct some spelling: i.e. Line 348, line 361, etc.

Material and Methods:

Line 89-90: Were preventive counseling sessions structured? How did you guarantee that all HCP provided the same information, at least, similar information?

Why did you provide topics to choose? Your aim was to explore women´s needs. It would have been more interesting to ask openly. As you describe in setting sections, this research seems to be a quasi-experimental study with an intervention about lifestyle choice by women among a group provided by the research team.

How many people refused to participate or dropped out? Reasons?.

Line 111-112: I consider more appropriate to include the study design before setting. This section should begin with the type of study carried out.

Actually, this paper assesses women´s opinion about a program about counseling, but not ask women about what had included in that program. If this happened, please, state it.

Results:

In 3.2.2 healthcare providers’ perspectives, none verbatim is included. Please, support ideas that you mention with verbatim.

Limitations: Authors should discuss that the research was developed during the pandemic so data should be interpreted under this umbrella.

Discussion.

This section should be in line with the results. However, the authors discuss new ideas. The authors do not discuss about women and hcp perspectives comparing them, and contrasting with other authors.

Conclusions: Conclusions are not in line with the results. The authors conclude that the intervention is effective, but this was not the aim of this research or paper. On the other hand, the kind of intervention was not assessed so it is not possible to state it in conclusions section.

In general lines, the paper provides interesting information, but aim, methods, results and conclusions should be better connected.  

Round 2

Reviewer 4 Report

The paper has been improved and now is suitable for publication.